# Accuracy to Predict the Onset of Calving in Dairy Farms by Using Different Precision Livestock Farming Devices

**DOI:** 10.3390/ani12152006

**Published:** 2022-08-08

**Authors:** Ottó Szenci

**Affiliations:** Department of Obstetrics and Food Animal Medicine Clinic, University of Veterinary Medicine Budapest, H-2225 Ullo Dora-major, Hungary; szenci.otto@univet.hu

**Keywords:** dairy cow, predicting of calving, precision livestock farming devices, perinatal mortality, dystocia

## Abstract

**Simple Summary:**

If the onset of calving can be accurately detected as well as appropriate calving assistance can be performed on a dairy farm, at that time, the prevalence of dystocia, stillbirth, vaginal laceration, retained fetal membranes, and consequent clinical metritis/endometritis can be decreased significantly. Therefore, in order to reduce these losses, our primary task must be to predict the onset of calving accurately and provide timely and professional calving assistance. This review focuses on the diagnostic possibilities and limitations of detecting the onset calving in the field.

**Abstract:**

Besides traditional methods such as evaluation of the external preparatory and behavioral signs, which even presently are widely used also in large dairy farms, there are several new possibilities such as measuring body (intravaginal, ventral tail-base surface, ear surface, or reticulo-ruminal) temperature, detecting behavioral signs (rumination, eating, activity, tail raising) or detecting the expulsion of the device inserted into the vagina or fixed to the skin of the vulva when allantochorion appears in the vulva to predict the onset of the second stage of calving. Presently none of the single sensors or a combination of sensors can predict the onset of calving with acceptable accuracy. At the same time, with the exception of the iVET^®^ birth monitoring system, not only the imminent onset of calving could be predicted with high accuracy, but a significantly lower prevalence rate of dystocia, stillbirth, retained fetal membranes, uterine diseases/clinical metritis could be reached while calving-to-conception interval was significantly shorter compared with the control groups. These results may confirm the use of these devices in dairy farms by allowing appropriate intervention during calving when needed. In this way, we can reduce the negative effect of dystocia on calves and their dams and improve their welfare.

## 1. Introduction

The profitability of cattle breeding is greatly influenced by the rate at which calves are born alive and reared to adulthood. Despite the speedy developments in animal breeding, perinatal mortality in Holstein-Friesian heifers and cows in different countries is still very high (3.5 to 8%) [1], and constitutes approximately half of the total calf losses [2,3,4,5,6]. Perinatal mortality (stillbirth) is defined as the death of a mature fetal calf after at least 260 days of gestation during calving or in the first 24 to 48 h of postnatal life [4,7].

During the last decades, there was a trend of increasing rates of stillbirths, especially in Holstein-Friesian (HF) heifers. In the Swedish HF-heifer population stillbirth rate has risen from 4% to 11% [8], while others reported that at first calving, around 10% of the calves were born dead or died on the first day [9,10,11]. In the Netherlands, the stillbirth rate for heifers was reported to be 12.2% in 1999 [12], and in the USA, it was 13.2% in 1996 [13]. It is essential to mention that instead of an increasing rate, recent studies show static or a declining trend in the stillbirth rate [14].

These figures emphasize the importance of examining the causal factors of perinatal mortality. The proximate cause-of-death (PCOD) with a non-infectious etiology is likely multifactorial. Still, most calves may die due to direct and indirect asphyxia because, in 73 to 75% of the calves that died in the perinatal period, no pathological changes were detected [15,16]. In other studies, asphyxia in calves dying perinatally was 58.3% [17] and 44.7% [18], respectively. According to recent necropsy studies, Mee [19] reported that the prevalence of anoxia was highly variable (~5 to ~80%) while the combined diagnosis rate with dystocia ranged between ~20 and ~45%. In a recent study, the ultimate cause-of-death (PCOD) with an infectious etiology was 34%, of which *Coxiella burnetii* was the most frequently detected pathogen [18]. This finding calls attention to the importance of monitoring the contagious etiology of perinatal mortality in those farms where the prevalence of stillbirth is high.

Due to the fact that the majority of the fetuses to be born can be lost because of asphyxia developing during calving, therefore, it is essential to detect the onset of calving accurately and the ability to distinguish between eutocia and potential dystocia [20] as well as to provide appropriately timed obstetrical assistance (70 min after the appearance of the amniotic sac or 65 min after the appearance of fetal hooves in the vulva: [21]) if it is needed. Inappropriately timed obstetrical assistance may lead to the high prevalence of dystocia, impairs postpartum health of the dam, and poses a potential risk to newborn calf survival [22]. Therefore, for smaller dairy farms, the ‘‘two feet–two hours’’ rule-of-thumb was suggested to decrease the use of calf pullers [23]. Qualifications of employed calving assistants, changes in the calving supervision during working shifts, or low-level surveillance of calvings, especially during nights and weekends or bank holidays, especially in large dairy farms, may contribute to the significant increase in stillbirth rate [4,24,25].

There are some positive results in American [26] and Canadian Holstein-Friesian dairy farms [27] with lower prevalence rates (<2%) of stillbirth; however, it is important to mention that sterile obstetrical lubricant was applied liberally to the dam’s birth canal around the fetus before performing the examination and providing obstetrical assistance.

All of this draws attention to the importance of accurately predicting the onset of calvings. Since the onset of calving cannot be detected accurately by observing behavioral and clinical signs of impending parturition, especially in large dairy farms, therefore, there has been a keen interest in the use of precision livestock farming devices (PLF) to predict calvings, which is also indicated by a large number of recently published scientific reviews [28,29,30,31,32,33,34] and meta-analyses [35,36,37,38] compiled according to different aspects.

The aim of the present review is to focus on the accuracy of predicting the onset of calving in dairy cows by using different PLF devices to decrease the prevalence of delayed calving assistance and the consequent stillbirth.

## 2. Prediction of Calving by Evaluating the External Preparatory Signs for Calving in Dairy Cows

For the determination of impendent parturition, the following clinical signs can be evaluated: udder starting to fill out, udder highly distended, udder edema, leaking of colostrum, swelling of the vulva, discharge of mucus from the vulva, and relaxation of pelvic ligaments. According to Berglund et al. [39], the general pattern for impendent calving is an enlargement of the udder starting on average 1–2 weeks before calving. Enlargement of the vulva starts at about the same time but is rather variable. The pelvic ligaments start to relax on average for one week before calving. The udder is highly extended 1–2 days before calving, the average time is 36 h before the birth of the calf, and in 75% of calvings, the udder fills well within the last 48 h. Relaxation of the pelvic ligaments (85.2%) and udder distension/licking of colostrum (75.6%) were the most reliable and useful signs to predict calving within 12 h. At the same time, no differences between breeds were found regarding preparation for calving [39].

A parturition score (PS) was developed (Table 1) by evaluating seven clinical signs (broad pelvic ligaments relaxation, vaginal secretion, udder hyperplasia, udder edema, teat filling, tail relaxation, and vulva edema) for field conditions to predict either calving or no calving within 12 h [40]. A threshold of 4 PS points was identified below, and calving within the next 12 h could be ruled out with a probability of 99.3% in cows and 95.5% in heifers (98.6% for cows and heifers), respectively. In contrast, the probability of predicting calving within 12 h in heifers and cows was 14.9%. It was also confirmed that changes in clinical signs during the last days of the preparatory stage were less informative in heifers than in cows, and relaxation of the broad pelvic ligaments and filling of the teats gave the best values for predicting either calving or no calving within 12 h [40].

By measuring the relaxation in pelvic ligaments /”One scale has to be kept firm exactly parallel to the ligament between the sacrum and the tuber ischii, and the other scale has to be erected perpendicularly to the first scale with the bottom just touching the ligament, and the depth can be measured in the second scale from the point where it touches the ligament to the point where it touches the first scale”/, calving within 24 h can be predicted with high accuracy (93.9%) as suggested by Shah et al. [41].

## 3. Prediction of Calving by Measuring Body Temperature in Dairy Cows

It has been shown for a long time that there is a pre-calving decrease (0.56 to 0.89 °C/1 to 1.6 F) in rectal [42,43] or vaginal temperature [44,45] before calving. Ewbank [46] found that healthy cows, even when exhibiting external signs of imminent parturition, such as mammary distension, relaxation of pelvic ligaments, and vulval enlargement, were unlikely to calve within the succeeding 12 h if their rectal temperature was above 38.8 °C. The accuracy of predicting calving by detecting temperature drop was only 43.5% [47], while a decrease in rectal temperature measured in the morning (0730 h) or in the evening (1700 h) over 24 h of ≥0.3 °C could predict calving within 24 h, with sensitivity from 44 to 69% and specificity from 86 to 88% [48].

It is important to mention that body temperature in cattle exhibits a circadian rhythm with a minimum temperature in the morning and a maximum temperature in the late afternoon [49,50]. At the same time, windy and rainy weather conditions [51], heat stress [52], and the effectiveness of cooling methods [53] may influence body temperature and the pattern of the circadian rhythm.

Using different temperature loggers has only recently made it possible to measure temperature continuously in the vagina, tail base, ear, and reticulo-rumen by this way, this diagnostic tool can be used for calving prediction 24 h before calving.

### 3.1. Vaginal Temperature (VT)

By measuring the vaginal temperature three times a day before calving in eight cows, Porterfield and Olson [44] reported that a drop in vaginal temperature could also be used to predict calving in over 50% of the cows, while in the remaining cows, the temperatures fluctuated so much that it was impossible to predict calving accurately. Since then, several reports have confirmed the pre-calving decrease in vaginal temperature by using sensors inserted into the vagina after attaching to a modified controlled internal drug release device at least six days before the expected time of calving retrospectively, as temperature data could be downloaded only after calving [48]. In this way, the optimal cut-off points of decrease in vaginal temperature (≥0.3 °C) one day before calving could be determined [48,54]. Due to significant diurnal variations (up to 0.5 °C) also, in vaginal temperatures, at least two temperature measurements are needed on a daily basis which makes temperature measurements impractical for calving prediction without converting them into automated signals [48,54].

According to Burfeind et al. [48], vaginal temperatures were continuously measured by temperature loggers inserted into the vagina about six days before expected calving. A decrease in the vaginal temperature of ≥0.3 °C over 24 h could predict calving within 24 h, with sensitivity ranging from 62 to 71% and specificity ranging from 81 to 87%, respectively. Similarly, a decrease in rectal temperature measured at 0730 h of ≥0.3 °C could predict calving within 24 h, with sensitivity from 44 to 69% and specificity from 86 to 88%.

Several remote devices are available for dairy farmers to record decreases in vaginal temperatures for the prediction of the onset of calving (Table 2). 

However, only a few authors reported on changes in vaginal temperature around calving in dairy cows based on Vel’Phone thermometers [55,57,61,62]. The sensitivity of receiving the “possible calving in 48 h” SMS message was 40% [55], while the sensitivity of the “expected calving in 48 h” SMS message was 82.9%, respectively. In contrast, Choukeir et al. [56] reported somewhat lower sensitivity results for possible and expected calvings in 48 h SMS messages (21.1% and 62.4%, respectively), while the positive predictive values of the SMS messages were 10.3% and 75%, respectively. Sakatani et al. [57] used another temperature sensor that recorded the vaginal temperature. An alert (Alert 1) was issued when the temperature fell below the threshold (Alert 1) and when the sensor reached the ambient temperature after falling out of the dam’s vagina with the rupture of the allantochorionic sac (Alert 2).

To increase the accuracy of measuring the vaginal temperature, Ricci et al. [63] have suggested using an intravaginal temperature of 38.2 °C as a cut-off value to predict calving within 24 h (vaginal temperature decreased from 38.65 °C to 38.12 °C between 48 and 60 h and 0 to 12 h before calving, respectively) because it can be more accurate (sensitivity: 86% vs. 66%) than a 0.21 °C decrease during the last 24 h before calving. Choukeir et al. [56] found similar values in dairy cows because the mean temperature of 0 to 6 h before calving was 38.19 °C; however, this suggestion must be confirmed.

According to Lammoglia et al. [64], vaginal temperatures were not affected by the gender of the calf, and there was no diurnal variation in body temperature from 48 to 8 h before calving in beef cows. Ricci et al. [63] reported that parity, dystocia, season, and length of gestation did not affect the vaginal temperature from 60 h before and up to calving, while Choukeir et al. [56] found that the vaginal temperature of dairy cows was significantly affected by parity, season (summer vs. autumn), time of day (8 a.m. vs. 8 p.m.), and the 6 h time intervals, whereas gender, birth weight of the calf, twinning, gestation length, fetal presentation, dystocia, and occurrence of retained fetal membranes did not affect it significantly. These results can be explained by a diurnal rhythm (up to 0.5 °C) in the vaginal temperature during the last 120 h before calving [48,54], while others could not confirm this pre-calving diurnal variation [63,64]. 

### 3.2. Ventral Tail Base Skin Temperature (TBST)

The ventral tail base surface temperature (TBST) can be measured using a tail-attached wireless sensor (Table 2) as described previously [58,59,65]. TBST is classified as peripheral temperature, which depends on the core temperature, environmental conditions, and peripheral blood system regulation. Environmental conditions such as ambient temperature, humidity, wind, sun, shade, and air movements may have a strong impact on body temperature [66].

TBST can be approximately 1.0 °C lower than VT throughout the prepartum days, and it shows a significant correlation (r = 0.56) with VT [65]. To exclude the effect of the circadian rhythm on TBST values, Miura et al. [67] suggested expressing their changes in residual TBST (rTBST = actual hourly TBST − mean TBST for the same hour on the previous 3 days). The general pattern of change, the points where the decrease starts, and the degree of decrease in both TBST and VT before parturition were almost identical [58,65]. At the same time, rTBST showed a biphasic decrease pattern [58,65]. An ambient temperature-independent (three ambient temperature groups: <15 °C, ≥15 °C to <25 °C, and ≥25 °C were examined) gradual decrease occurred from around 36 to 16 h before calving, and an ambient temperature-dependent sharp decrease occurred from around 6 h before until calving. It is important to mention that the accuracy of predicting calving tended to be lower at lower ambient temperatures. At the same time, the rearing condition (free-stall barn vs. tie-stall barn) in dairy cattle did not influence the accuracy of predicting calving [59].

A further advantage of measuring the ventral tail base surface temperature in the field is that after calving, temperature measurements can be continued without interruption, and in this way, it can be used to predict different diseases associated with body temperature change (e.g., retained fetal membranes, metritis, mastitis) in a timely manner [65].

### 3.3. Ear Temperature

Ear surface temperature is greatly dependent on the ambient temperature; therefore, Stevenson [68] suggested to group the animals in the hotter (between May and September) and the colder months (between October and April) according to the hourly measured ear-surface temperatures during days 230 and 239 of gestation into two median temperature groups in each season: high temperatures (range of 33.67–38.89 °C, mean ± SEM = 32.9 ± 0.2 °C; medium-high temperatures (25.06–31.66 °C, 29.4 ± 0.2 °C); medium-low temperatures (ML, 17.82–25.00 °C, 21.1 ± 0.2 °C) and low temperatures (0.70 to 17.80 °C, 13.4 ± 0.2 °C), respectively. Daily prepartum ear-surface temperatures were fairly constant in each temperature group and linear during the last ten days of gestation until the last 24 h before calving when temperature decreased abruptly, which were more expressed in the colder months. However, further studies are needed to evaluate the accuracy of ear-temperature measurements in the field.

### 3.4. Reticulo-Rumen Temperature

Orally administered temperature-sensing reticulo-rumen bolus has a temperature sensor that can measure reticulo-rumen temperature (Trr) every hour and store up to 12 readings. The telemetric system is equipped with two antennas with a reach capacity of 90 m and locates within the maternity pen. Therefore, complete records required that an animal is within the area of reach of the antennas between 2 and 3 times per day. From the antennas, readings are sent to a computer, which can store the Trr data and can be used for evaluation [60].

Water intake depending on the temperature and its volume may cause a transient reduction in reticulo-rumen temperature; therefore, Boehmer et al. [69] suggested removing the Trr temperatures below 37.7 °C. It is important to mention that the Trr due to the heat production of reticulo-rumen microorganisms is around 0.5 °C higher than the body temperature [70], while this difference is due to lower nutritional intake in beef cows being redundant [71]. According to Costa et al. [60], a temperature change threshold of ≥−0.2 °C provided the best predictive performance compared with the ≥−0.3 °C and ≥−0.4 °C thresholds for all Holstein females (primiparous and parous: sensitivity 69%, specificity: 69%) when instead of 1 h window a 5 h window was used to evaluate the difference between the four previous days reading from the current reading (Table 2). According to Kim et al. [72], the reticulo-rumen temperature in Hanwoo (Bos taurus coreanae) cows can be lower by 0.5 °C from −24 h to −3 h before calving compared to 48 h before parturition.

## 4. Prediction of Calving by Evaluating the Behavioral Signs Using Different Sensors

According to previous examinations reviewed recently by Saint-Dizier and Chastant-Maillard [28] and Matamala et al. [31], behavioral changes on the actual day of calving compared with preceding days or 2 to 6 h before calving compared with preceding hours on the actual day of calving, i.e., within 24 h before calving such as lying time (during the final 2 h period), activity (steps and restlessness on the actual day of calving, or head turns and stamping in the final 2 h period), number of lying/standing position (during the actual day of calving with a peak in the last 2 h), isolation (during the actual day of calving), tail raising (during the final 2–4 h period), lateral lying position with head rested (during the final 4 h period), and abdominal contractions (during the final 4–8 h period with a peak at the last 2 h) used to be increased. While, feeding time (during the actual day of calving or the final 2–6 h period), drinking time (during the final 2 h period), dry matter intake (during the actual day of calving or the final 6 h period) ruminating time (during the actual day of calving or the final 4–6 h period) and neck activity (18 h before calving) used to be decreased which can be used to predict calving. At the same time, others found no variation in feeding time, drinking time, or water intake, as reviewed by Saint-Dizier and Chastant-Maillard [28]. In a recent examination, Stevenson [68], using three-dimensional accelerometers in ear tags, found a significant decrease in eating and rumination time before calving independently from the season and a significant increase in resting and active time before calving, which was significantly influenced by the season. 

### 4.1. Accuracy to Predict Calving by Evaluating the Behavioral Signs of Imminent Calving by Using a Single Sensor

Rumiwatch noseband sensors can detect rumination time, eating time, and other activity, i.e., non-ingestive related behaviors and the frequencies of ruminating boluses, ruminating chews, ruminating chews per bolus, ruminating chews per minute, eating chews and other chews, i.e., non-ingestive related jaw movements (Table 3); however, none of these parameters can be used accurately to predict calving 1 h before calving or just before calving (not given in Table 3). Even combining the individual parameters did not help to increase the accuracy of the forecast either [73].

Rutten et al. [74] evaluated the data of a single sensor in an ear tag (n = 400) which was able to synthesize cumulative activity, rumination activity, feeding activity, and temperature on an hourly basis. Two logit models were developed: a model with the expected calving date as an independent variable and a model with additional independent variables based on sensor data. The areas under the curves of the receiver operating characteristic were 0.885 and 0.929 for these models, respectively. The model with the expected calving date was evaluated on a daily basis and only had a sensitivity of 9.1% (specificity: 99.3%), whereas the model with additional sensor data had a sensitivity of 36.4% (specificity: 98.9%). At the same time, the model with the expected calving date and sensor data had a sensitivity of 21.2% at a 1 h time window, 42.4% at a 3 h time window, 48.5% at a 6 h time window, 51.5% at a 12 h time window, while the specificity changed from 99.1 to 99.4%, respectively. Similar results were reported by Ouellet et al. [54] and Krieger et al. [75]. This indicates that prediction of the specific hour in which calving started is not possible with a high accuracy when ear tag sensors are used.

Similarly, using hind leg accelerometers or neck collar accelerometers and a microphone cannot increase the accuracy of predicting calving at different time points. The sensitivity in each case was lower than 80%, while the specificity was less than 92.9%. 

Tail raising and behavioral changes can be detected by a tail-mounted tri-axial accelerometer attachment to a cow tail [80,84], a tail-mounted accelerometer, and other gravitational measurement devices [85], a tail-mounted inclinometer/Moocall/sensor [81], or a tail-mounted inclinometer and an accelerometer/Moocall/sensor [82]. Tail-raising will be increased in the final 2–4 h period compared with preceding hours on the actual day of calving, i.e., within 24 h before calving [28,85]. In this way, this can be used to predict the onset of calving. There are several tail-mounted sensors in the market; however, only Moocall sensors were used for scientific evaluations. Moocall sensors can generate two types of signals depending on the tail activities: SMS type 1 alert (SMS1) if enhanced activity can be registered over one hour and SMS type 2 alert (SMS2) if a high activity will be continued in the consecutive hour before imminent calving. Giaretta et al. [82] reported that the cows could well tolerate the Moocall sensors; however, they examined only 12 cows. In contrast, Voß et al. [81] reported that in 31 animals, the sensor was removed because the tail was swollen or painful. Sensors continuously remained on the tail (i.e., within 3 cm of the initial attachment position) after initial attachment until the onset of calving in only 13.9% of animals (n = 25) and had to be reattached until a calving event occurred (51.6%). Horváth et al. [62] also reported that some cows did not tolerate the device well, and the average number of SMS messages per animal was too much (mean ± SD: 12.7 ± 15.2), and the majority of these messages were false positive. In agreement with others [86], Mee et al. [85] reported that cows dislodged some devices from each other’s tails by licking and chewing, and after a period of 3–4 days, edema of the tail above and below the device developed in some cows which problems were solved when the weight of the device was decreased (from 133 g to 50 g) and the stronger wrap was used for fixing the device. Although the first report was very promising, with a sensitivity of 100% and specificity of 95% within 24 h of calving [82], however, this accuracy could not be reached in a larger study by Voß et al. [81] because the sensitivity of the test varied from 19 to 75% (the higher value 24 h before calving) and specificity from 63 to 96% (the higher value 1 h before calving). Similar results were reported by Miller et al. [80] when the sensitivity (78.6%) and specificity (83.5%) of their tail-mounted sensor were evaluated in a 5 h window before calving. Horváth et al. [62] also emphasized that Moocall sensors could not inform about the exact time of calving, but they were helpful in optimizing worker efficiency.

By using precision cow monitoring technologies such as image processing techniques, visual information for the assessment of calving behaviors (lying, standing, holding up the tail, turning the head to the side) can be provided, and by using an integrated hidden Markov model, calving can be predicted with high sensitivity (91.5%) in heifers within 3 h before calving. However, only 15 cows were involved in the experiment [83].

### 4.2. Accuracy to Predict Calving by Evaluating the Behavioral Signs of Imminent Calving by Using a Combination of Sensors

Localization sensors and neck- and leg-mounted accelerometers (Table 4) were used to evaluate these sensors to predict the onset of calving in 13 pregnant cows [79]. These sensors made it possible to evaluate lying time, ruminating time, number of steps, and travel distance before calving. The sensitivity to predict calving within 24 h using localization sensors and neck- or leg-mounted sensors changed from 74 to 78%. A somewhat lower result (68%) was reached when the neck- and leg-mounted sensors were used together. However, when the neck- and leg-mounted sensors were used together with the localization sensors, the sensitivity of these sensors became 85%. At the same time, the specificity of the sensors changed from 97 to 98%. Similar results were reached 12 h and 8 h before calving, while the sensitivity of using the different sensors together to predict calving 4 h and 2 h before calving decreased to 54–69% and 42–63%, respectively. At the same time, the specificity of the sensors was similar (95 to 97%) to 24 h values.

Ouellet et al. [54] measured rumination time (ear tag), vaginal temperature (temperature data logger), and lying behaviors (right hind leg) and reached 77% sensitivity and 77% specificity to predict calving within the next 24 h. 

Fadul et al. [87] measured ruminating time, ruminating chews, boluses, and other activities not related to ruminating, feed intake or drinking activity (noseband sensor), and lying bouts (3D accelerometer). Sensitivity of 88.9% and 85% and specificity of 93.3% and 74% for primiparous and multiparous dairy cows were reported to predict calving within three hours before calving.

Borchers et al. [77] used a neck and left hind leg sensor to measure neck activity and rumination as well as the number of steps, time spent lying, time spent standing (inverse of time spent lying), number of transitions from standing to lying (lying bouts), and proprietary total motion variable in 15 min periods. The sensitivity and specificity to predict calving within an 8 h period by using neural network analysis were 82.8% and 80.4%, respectively, while it was 100% when calving was predicted within 24 h.

Liseune et al. [89] reported that by using sequential deep learning algorithms for evaluating behavioral activities such as eating, ruminating, walking, and lying, the moment of parturition could be predicted better than by using a more traditional machine learning algorithm because 12 h before calving the sensitivity at 0.3 thresholds was 98%, while the specificity was only 15%.

It seems that the accuracy of predicting the onset of calving can be increased by using different sensors together; however, further studies are needed to confirm which combination of the sensors would be the most accurate. Similarly, it is also very important to select a correct machine learning technique to evaluate our results because Quddus et al. [88] were able to reach a high accuracy (sensitivity: 100, specificity: 98.9%) by using the neural network analysis, to predict calving in dairy buffaloes within 24 h before calving. Keceli et al. [90] also emphasized the importance of selecting the right algorithm to evaluate the activity and behavioral data 24 h before calving because by using the Bi-LSTM (bi-directional long short-term memory) network-based prediction model, the sensitivity and specificity became 100%, while by using the LSTM model the sensitivity and specificity were only 86% and 98%, respectively.

## 5. Detection of the Expulsion of Sensors during Appearing Allantochorion at the Beginning of 2nd Stage of Labor in Dairy Cows

Presently there are two possibilities to detect the expulsion of the sensors during appearing allantochorion at the beginning of the second stage of labor in a dairy farm: namely using vulvar magnetic or intravaginal sensors (Table 5).

### 5.1. Vulval Magnetic Sensors

A vulval magnetic device was originally developed for horses to predict the onset of foaling to give immediate obstetrical assistance if needed. This is an invasive method and needs a veterinarian to suture the transmitter and the magnetic device to the vulval skins. Appearing allantochorion separates the vulval lips, and the magnetic device moves out of the sensing field of the reed switch, turning on the transmitter and allowing communication with a remote receiver. The onset of calving can be predicted with high accuracy (sensitivity: 100%, positive predictive value: 95 to 100%). It was reported that false alarms (3 of 58) were caused by the friction of the animal against the fence of the barn, with the consequently accidental separation of the two parts of the transceiver [92]. Similarly, movement of the tail may also cause dislodging of the device, and urine or feces may cover the device. Fixing the device with tissue glues does not currently seem possible for the duration necessary for calf alert requirements [99].

### 5.2. Intravaginal Sensors

There are also two types of intravaginal sensors: namely physical (light) and temperature sensors. Each of them has to be inserted aseptically into the vagina until contact with the external cervical os by a gloved hand [94,100] or a vaginal applicator [56,57,61].

The vaginal probe developed by Palombi et al. [94] consists of an anchoring base and a cylindrical bin. The anchoring base is fin-shaped in order to secure the device to the vaginal wall, and the bin contains physical sensors. In the case of a Vel’Phone, depending on the size of the cow, two appendage kits are available for heifers (turquoise) and multiparous cows (white), which prevent their loss [61]. It is important to mention that after repeated uses of the appendage kits, they may become too rigid, so it is advisable to replace them as recommended by the manufacturer. iVET^®^ birth monitoring system has a T-shape [98], while the Gyuon-kei temperature sensor has a stopper [57] which can prevent its loss. The intravaginal sensors will be expelled from the birth canal when the allantochorionic sac enters it. The light sensor can generate an output even in case of scarce brightness, such as at dusk or at night, while the temperature sensor generates an alert if the difference between the dam’s temperature and the external environment is enough. Heat stress may prevent to generating of an alert; however, this was not observed by us in continental weather conditions.

With the exception of the iVET^®^ birth monitoring system, none of the physical and temperature sensors induced any pathological clinical signs with the exception of a minor discomfort shown by some heifers [56,61]. When the intravaginal device remained inside the vaginal canal for two consecutive weeks, Palombi et al. [94] observed no adverse effects, and the animals did not exhibit any discomfort or vaginal discharge. In contrast, Henningsen et al. [98] and Marien et al. [100] reported that the iVET^®^ birth monitoring system caused a significantly higher number of injuries and extreme calving difficulties. The injuries were more severe, the healing progressed more slowly, and these animals developed endometritis significantly more frequently than the control group. Maybe the size of the monitoring system caused this discomfort in heifers. Stagnation of calving or premature rupture of the allantochorionic sac in some heifers also occurred; therefore, a smaller version has been developed for heifers. However, the benefit of these changes has not been confirmed yet. Recurrent vaginal prolapse may abort the physical sensor, as found in a buffalo heifer, while poor carrier network coverage may also negatively affect the phone’s signal quality [101].

In practical circumstances, with the exception of two reports, the intravaginal sensors could be used with high accuracy. The sensitivity changed between 97.2 and 100%, while the positive predictive value was between 92.8 and 100% (Table 5). When OraNasco^®^ physical sensors were used, the lower sensitivity was caused that no alarm was received at the beginning of calving due to the failure of the GSM network in the area that occurred during the study period [96]. In contrast, when the iVET^®^ birth monitoring system was used, perhaps the size of the device can be blamed for the lowest sensitivity [97,98].

Except for one study, not only the second stage of calving was predicted with high accuracy but significantly lower prevalence rate of dystocia [61], stillbirth [61,91,94,96,98] (only numerically lower), retained fetal membranes [61,91,94], uterine diseases/clinical metritis [61,91,94] were reported. At the same time, the calving-to-conception interval was significantly shorter [91,94,96] compared with the control groups and greatly contributed to the welfare of these animals [1,36,37].

## 6. Future Perspectives

Presently it seems that we cannot predict accurately (sensitivity: >95%) the onset of calving in dairy cows by evaluating the external preparatory clinical signs for the onset of calving and by using different sensors to detect the decrease in core temperature (vaginal, ventral-tail-base surface, ear-surface, or reticulo-rumen temperature) or evaluating behavioral signs by using a single or a combination of sensors. However, by using a combination of sensors, we can increase the accuracy of predicting the onset of calving (Table 4). At the same time, Santegoeds [102] emphasized by examining 3000 calving cows in 19 commercial Dutch dairy farms that the accuracy of predicting calving in a commercial setting based on behavioral variables (steps, head movements sharp up, stand-ups, eating, lying, standing, walking,) measured by smart tags (neck and leg) could not be improved. Górriz-Martin et al. [86] reported that we might increase the accuracy of predicting the onset of calving by detecting the external preparatory clinical signs for calving and using a sensor on the tail; however, it would be labor- and time-consuming in a dairy farm.

At the same time, the onset of calving can be predicted accurately (sensitivity: >97%) if we insert devices into the vagina some days before the expected calving, which will be removed from the vagina during appearing of the allantochorionic sac in the vulva just at the beginning of the second stage of labor (Table 5). It is important to mention that the device must not irritate the vagina and cause discomfort to the animals, such as in the case of the iVET^®^ device. Similarly, the vulval magnetic sensors can be used with high accuracy. However, this is an invasive method because it needs a veterinarian to suture the transmitter and the magnetic device to the vulval skins, which limits their use in large dairy farms. With a newly discovered sensor or a correct combination of the sensors or just improving the algorithms used for evaluating the sensor data, we can increase the accuracy of detecting the onset of calving to further decrease the delayed obstetrical assistances which may cause dystocia and stillbirth. It was recently confirmed that if we can do the appropriately timed obstetrical assistance after the onset of the second stage of labor, we can significantly decrease the prevalence of dystocia, stillbirth, retained fetal membranes, and vulvovaginal lacerations compared with the inappropriately timed obstetrical assistance [22]. By predicting the onset of calving, we can decrease not only the prevalence of dystocia and stillbirth but also the prevalence of retained fetal membranes and clinical metritis [61,91,94], and we can improve animal welfare [1,36,37].

Another aim to use different sensors in a dairy farm would be to predict dystocia before calving. Matamala et al. [31] have recently reviewed behavioral changes (feed intake, rumination time, lying bouts, tail elevation) associated with dystocia and concluded that they seem to be promising in the early detection of cows with a higher risk of dystocia. However, most of the examined studies used small sample sizes (8 to 12 cows); therefore, in agreement with Chang et al. [35,103], further research is needed to assess differences among parity, breeds, and different housing conditions, including pasture-based systems in larger data sets. At the same time, Cavendish et al. [104], examining 35 dairy cows by video surveillance, could not detect any differences in behavior between assisted and unassisted cows, while Mammi et al. [105] reported that rumination time can be shorter in case of dystocia. According to Kovács et al. [106], the decrease in reticulo-rumen temperature may occur 12 h earlier in cows with dystocia than in eutocic cows.

## 7. Conclusions

One of the most important management activities needed to pursue during the periparturient period is to provide obstetrical assistance at an appropriate time after detecting the onset of the second stage of labor to decrease the prevalence of dystocia and stillbirth and to improve animal welfare. There are several diagnostic methods to detect the onset of calving available for the dairy farms, such as controlling the external preparatory clinical signs, measuring the body temperature by different sensors, evaluating the behavioral signs by using a single or a combination of sensors, or inserting a sensor into the vagina or suturing it to the vulval skin to detect the second stage of labor when the allantochorionic sac was appearing in the vulva. The advantages and disadvantages of the different diagnostic methods were discussed in order to be able to select the most accurate method for the diagnosis of the onset of calving on a dairy farm.

## Figures and Tables

**Table 1 animals-12-02006-t001:** Use of different clinical signs during the preparatory stage of cattle for a parturition scoring system [35].

Clinical Signs	Parturition Score			
	0	1	2	3
Relaxation of the broad pelvic ligaments	Firm, no-marginal relaxation 0 to 20%	Mildly softened up to 50%	Totally softened, but palpable up to 100%	Totally softened, not palpable 100%
Secretion of vaginal mucous ^a^	None	Slight <10 cm long diameter <1 cm	Moderate >10 cm long diameter <1 cm	Extensive >10 cm long diameter >1 cm
Physiological hyperplasia of the udder	Empty, small palpable	Slightly filled	Partially filled	Totally filled, enlarged, not palpable
Edema of the udder	None	On the base	Entire udder	Including the abdomen
Filling of the teats	Flaccid None	Slightly filled ∼25%	Moderately filled ∼50%	Completely filled ∼100%
Relaxation of the tail ^b^	No flexibility	45°∼90°	90°∼120°	120°∼180°
Edema of the vulva ^a^	Strongly folded, no Edema	Moderately folded, mild Edema	Mildly folded, moderate Edema	Not folded, high Edema, redness of inner mucosa

^a^ The tail has to be lifted to evaluate the vaginal mucous and edema of the vulva. ^b^ The relaxation of the tail is tested by flexing the last third of the tail. The degree of flexure without any defense reaction should be estimated.

**Table 2 animals-12-02006-t002:** Accuracy to predict calving by using different temperature loggers in dairy cows.

Sensor Type	Event	Device	Number of Animals	Time	Sensitivity (%)	Specificity (%)	References
Intravaginal temperature data logger	Vaginal temperature (<0.2 °C) ^a^	Minilog 8 (attached to CIDR) ^b^	85	48 h	70–80 ^c^	73–81 ^c^	Burfeind et al. [48]
				24 h	71–78 ^c^	71–79 ^c^	
	Vaginal temperature (0.1 °C/6 h at 24 h and 0.2 °C/6 h at 12 and 6 h)	Minilog II-t (attached to CIDR)	42	24 h	74	74	Ouellet et al. [54]
				12 h	69	69	
				6 h	68	67	
	Vaginal temperature	Vel’Phone	35	Predicting calving with 48 h SMS	82.9	-	Chanvallon et al. [55]
	Vaginal temperature	Vel’Phone	215	Predicting calving with 48 h SMS	62.4	-	Choukeir et al. [56]
	Vaginal temperature (0.3 °C)	Gyuonkei	44	Predicting calving by Alert 1	79.5	-	Sakatani et al. [57]
Tail temperature sensor	Ventral tail base surface temperature (0.36 °C warm, 0.28 °C cold season) ^d^	-	35	Calving within 24 h	80–89	89–91	Koyama et al. [58]
				Within 18 h	83–92	87–88	
				Within 12 h	84–90	82–85	
				Within 6 h	83–90	79–82	
	Ventral tail base surface temperature ^d^	-	108	Calving within 24 h	84.3	-	Higaki et al. [59]
Reticulo-rumen temperature	Temperature-sensing reticulo-rumen bolus (≤0.2 °C) ^a^	-	261	Calving within 24 h	69 ^e^, 69 ^f^	69 ^e^, 69 ^f^	Costa et al. [60]
				Within 12 h	69 ^e^, 70 ^f^	65 ^e^, 65 ^f^	

Sensitivity: proportion of positive events (occurrence of calving within the examined time period) correctly predicted by the test (calving correctly predicted/total calving events). Specificity: proportion of negative events (absence of calving within the examined time period) correctly diagnosed as being negative by the test (absence of calving correctly predicted/total of absence of calving). ^a^ Cut-off value. ^b^ CIDR: modified controlled internal drug release device without progesterone. ^c^ Values were evaluated in three different experiments. ^d^ Residual temperature = actual body surface temperature−mean body surface temperature for the same hour on the previous 3 days. ^e^ Average of readings for 4 previous days using a 1 h window from the current reading. ^f^ Average of readings for 4 previous days using a 5 h window from the current reading.

**Table 3 animals-12-02006-t003:** Accuracy to predict calving by evaluating the different behavioural signs of imminent calving by using a single sensor in dairy cows.

Sensor Type	Event	Device	Number of Animals	Time	Sensitivity (%)	Specificity (%)	References
Noseband	Rumination time	RumiWatch (3D accelerometer)	24 (n = 11 and n = 13) ^a^	1 h	73.8	87.6	Zehner et al. [73]
	Eating time				27.7	89.6	
	Other activity time				91.7	48.7	
Ear	Activity, rumination, feeding, and temperature	SensOor Agis (3D accelerometer)	400	Hourly basis (12 h, 6 h ^b^, 3 h ^b^ and 1 h ^b^)	51.5	99.4	Rutten et al. [74]
				Daily basis	36.4	98.9	
Ear	Activity, rumination, and lying time	SMARTBOW (3D accelerometer)	444	Hourly basis (24 h ^b^, 12 h ^b^, 6 h ^b^, 3 h ^b^ and 1 h)	54	94.5	Krieger et al. [75]
Ear	Rumination time	SensOor Agis (3D accelerometer)	42	Hourly basis (22 h, 12 h and 6 h)	51–63	51–63	Ouellet et al. [54]
Right hind leg	Lying bouts	Onset Pendant G data logger			39–67	27–63	
	Lying time				48–57	47–57	
Hind leg	Standing and lying time, standing bouts	Gemini Datalogger (accelerometer)	101	24 h period	77.8	77.8	Proudfoot et al. [76]
-	Dry matter intake	Insentec electronic feed and water intake system			72.7	81.8	
	Feeding time				63.6	54.6	
	Water intake				81.8	54.6	
Left hind leg	No. of steps, total motion, lying time and lying bouts	IceQube (3D accelerometer)	53	8 h period	65.5–79.3 ^c^	78.6–83.9 ^c^	Borchers et al. [77]
Neck collar	Neck activity and rumination	HR tag (3-axis accelerometer and a microphone)			58.6–79.3 ^c^	80.4–92.9 ^c^	
Neck collar	Neck activity and rumination	Hi Tag (3-axis accelerometer and a microphone)	27	24 h period	~70	~70	Clark et al. [78]
Neck collar	Ruminating, feeding, resting time	Neck-mounted accelerometer	25	Hourly basis (24 h, 12 h, 8 h, 4 h ^b^, and 2 h ^b^)	47–48	94–95	Benaissa et al. [79]
Right hind leg	Lying time, lying bouts, number of steps	Leg-mounted accelerometer			54–56	94–96	
Neck collar	Travelled distance, Time in cubicles, time in feeding zone, time in drinking zone	Localization node			55–58	93–96	
Neck collar	Rumination	Silent Herdsman collar (neck-mounted accelerometer)	110	5 h window	69.8	59.3	Miller et al. [80]
	Eating				59.3	61.7	
	Activity				66.7	62.3	
Tail	Tail raising	Tail-mounted tri-axial accelerometer			78.6	83.5	
Tail	Tail raising	Moocall (tail-mounted inclinometer)	118	Hourly basis (24 h, 12 h, 4 h, 2 h ^b^, and 1 h ^b^)	66–75	63–89	Voß et al. [81]
Tail	Tail raising	Moocall (tail-mounted inclinometer and accelerometer)	12	24 h	100	95	Giaretta et al. [82]
				3 h	95.2	71.4	
-	Lying, standing, holding up the tail, turning the head to the side	Camera (360-degree GV-FER5700 camera) + behavior integrated hidden Markov model	15 ^d^	<3 h	91.5	-	Sumi et al. [83]

Sensitivity: proportion of positive events (occurrence of calving within the examined time period) correctly predicted by the test (calving correctly predicted/total calving events). Specificity: proportion of negative events (absence of calving within the examined time period) correctly diagnosed as being negative by the test (absence of calving correctly predicted/total of absence of calving). ^a^ Sensitivity and specificity were calculated separately in the two groups. ^b^ Values were not given because they were lower than the given values. ^c^ Sensor data in addition to the expected calving date. ^d^ Only heifers were examined.

**Table 4 animals-12-02006-t004:** Accuracy to predict calving by evaluating the behavioural signs of imminent calving by using a combination of sensors.

Sensor Type	Event	Device	Number of Animals	Time	Sensitivity (%)	Specificity (%)	References
Ear tag Hind leg Vaginal temperature	Rumination time, Lying bouts, lying time, Vaginal temperature	SensOor (3D accelerometer), Onset Pendant G data logger, Minilog II-t	42	Hourly basis (24 h, 12 h, and 6 h)	68–77	68–77	Ouellet et al. [54]
Noseband Hind leg	Rumination time, chews, boluses and other activities, Lying bouts, time, walking time and other leg movement	Noseband sensor (Rumiwatch) 3D accelerometer	33	3 h period	88.9 (primiparous) 85 (multiparous)	93.3 74	Fadul et al. [87]
Neck Hind leg	Rumination time, neck activity, No. of steps, lying time, lying bouts	HR Tag (3-axis accelerometer and microphone), IceQube (3-axis accelerometer)	53	24 h period ^a^ 8 h period ^a^	100.0 82.8	86.8 80.4	Borchers et al. [77]
Neck Hind leg	Feeding, rumination Lying, standing, No. of steps, standing time	NEDAP logger NEDAP logger	40 ^b^	24 h period ^a^	100	98.9	Quddus et al. [88]
Neck Front leg	Eating, rumination and lying time No. of steps, standing, walking and lying time	Nedap Smarttag Neck sensor Nedap Smarttag Leg sensor	572	Hourly basis (24 h, 12 h, 6 h, 3 h ^c^, and 1 h ^c^) Threshold: 0.3	87–98	15–81	Liseune et al. [89]
Neck Hind leg Localization	Ruminating, feeding, resting time Lying time, lying bouts, no. of steps Travelled distance, time in cubicles, feeding zone and drinking zone	Neck-mounted accelerometer Leg-mounted accelerometer Localization node	25	Hourly basis (24 h, 12 h, 8 h, 4 h ^c^, and 2 h ^c^)	79–85	97–98	Benaissa et al. [79]
Neck Tail	Rumination, eating, activity Tail raising	Silent Herdsman collar (neck-mounted accelerometer), Tail-mounted (tri-axial accelerometer)	110	5 h period	79.2	81.3	Miller et al. [80]

Sensitivity: proportion of positive events (occurrence of calving within the examined time period) correctly predicted by the test (calving correctly predicted/total calving events). Specificity: proportion of negative events (absence of calving within the examined time period) correctly diagnosed as being negative by the test (absence of calving correctly predicted/total of absence of calving). ^a^ Neural network machine-learning techniques were used to predict caving. ^b^ Dairy buffaloes were examined. ^c^ Values were not given because they were lower than the given values.

**Table 5 animals-12-02006-t005:** Accuracy to detect the expulsion of the sensors during appearing of the allantochorion at the beginning of 2nd stage of calving in dairy cows.

Event	Sensor Type	Device	Number of Animals	Time	Sensitivity (%)	Positive Predictive Value (%)	References
Vulvar lips separation	Magnetic sensor	C6 birth control	80	0 h	100	100	Paulocci et al. [91]
		C6 birth control	53		100	95	Marchesi et al. [92]
		GPS-calving alarm	18		100	100	Calcante et al. [93]
Intravaginal device expulsion	Physical sensor	-	120		100	100	Palombi et al. [94]
		Patent	117		100	100	Crociati et al. [95]
		OraNasco^®^	83		86.3 ^a^	-	Crociati et al. [96]
	Temperature sensor	Vel’Phone^®^	35		100	100	Chanvallon et al. [55]
		Vel’Phone^®^	257		100	100	Choukeir et al. [56]
		Vel’Phone^®^	44		100	100	Horváth et al. [62]
		Gyuonkei (−0.3°C)	44		97.2	-	Sakatani et al. [57]
		iVET^®^	54		74.1	92.6	Dippon et al. [97]
		iVET^®^	167 ^b^		78	93	Henningsen et al. [98]

Sensitivity: proportion of positive events (occurrence of calving within the examined time period) correctly predicted by the test (calving correctly predicted/total calving events). Positive predictive value: proportion of positive events (occurrence of calving within the examined time period) correctly predicted by the test (calving correctly predicted/calving correctly and incorrectly predicted). ^a^ No alarm was received at the beginning of calving due to the failure of the GSM network in the area that occurred during the study period. ^b^ Only heifers were examined.

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
