# Peer review of "Accuracy to Predict the Onset of Calving in Dairy Farms by Using Different Precision Livestock Farming Devices"

_animals, 2022, doi:10.3390/ani12152006_

Round 1
Reviewer 1 Report
The paper entitled “Accuracy to predict the onset of calving in dairy farms by using different precision livestock farming devices” illustrates the latest methods for the identification of ready-to-calve cattle in order to improve the timing of obstetrical assistance and to reduce the negative consequences of dystocia on both the dam and the calf.
The review is well organized and accurate in any detail. Tables are complete and informative. References are correctly updated, including the latest papers.
In my opinion the manuscript is sound to be approved for publication; only few minor observations as described below:
Line 42: maybe the author could add more references for stillbirth rates, which are referred to latest years.
Line 127-130: I suggest to add the time interval (hours) for calving prediction
Line 174: I suggest to change “the presence of retained fetal membranes” with “occurrence of retained fetal membranes”.
Line 237-250: the sentence is maybe too long. I suggest placing at least one point, for example at line 246: "... be increased. Actually, feeding time…”
Line 285: “sensitivity” maybe you intended “specificity”?
Line 328: I suggest changing with: “…within 3 hours before calving. However only 15 cows were involved in the experiment [78].”
Line 353-355: just a clarification is needed with this sentence. More in detail, the intravaginal devices are expelled from the birth canal when the sacs or fetus enter the canal itself. Vulvar magnetic sensors are not expelled in this case, but are separated when the birth canal dilates. Please, I suggest to re-arrange the phrase accordingly.
Line 420-431 or 435-438: in these paragraphs the author describes both pros and cons of the different intravaginal devices for the identification of calf expulsion. Among adverse effects/defects, I suggest to mention that vaginal retention issues of the intravaginal device were observed in buffalo heifers with recurrent vaginal prolapses, as described in [Remote monitoring system as a tool for calving management in Mediterranean Buffalo heifers (Bubalus bubalis ). Emanuela Rossi, et al. Reproduction in domestic animals. Volume55, Issue12 Vol. 55, Issue 12 December 2020 Pages 1803-1807. https://doi.org/10.1111/rda.13805]. Thus, pathological conditions which can influence the functionality of vaginal tissues should be accounted for and discussed, when considering the use of those devices as calving alarms.
Line 446-458: in this paragraph the author outlines the potential use of authomatic sensors for calving prediction; I only suggest to summarize which time interval is achieved for the prediction of calving with those systems.
Line 465: “invasive” instead of “invasion”?
Author Response
Responses to Reviewer Comments:
The author is grateful for the efforts of Reviewers in the evaluation of his manuscript. I appreciate your time spent with the review. Before answering the most important concerns, let me thank you for your valuable comments on the paper. I feel that Reviewers’ comments and recommendations were reasonable, and I tried to take them into account as far as possible while improving the manuscript. In my opinion, the activities of the reviewers have contributed significantly to the improvement of the quality of my paper. As you will see, I have made all the corrections required.
Reviewer I Comments
The paper entitled “Accuracy to predict the onset of calving in dairy farms by using different precision livestock farming devices” illustrates the latest methods for the identification of ready-to-calve cattle in order to improve the timing of obstetrical assistance and to reduce the negative consequences of dystocia on both the dam and the calf.
The review is well organized and accurate in any detail. Tables are complete and informative. References are correctly updated, including the latest papers.
In my opinion the manuscript is sound to be approved for publication; only few minor observations as described below:
AU: The author would like to thank Reviewer I for finding merit in the manuscript.
Rew#1: Line 42: maybe the author could add more references for stillbirth rates, which are referred to latest years.
AU: The following was added to the text: It is essential to mention that instead of an increasing rate, recent studies show static or a declining trend in the stillbirth rate [14].
Rew#1: Line 127-130: I suggest to add the time interval (hours) for calving prediction
AU: It was added: this diagnostic tool can be used for calving prediction 24 h before calving.
Rew#1: Line 174: I suggest to change “the presence of retained fetal membranes” with “occurrence of retained fetal membranes”.
AU: It was changed: occurrence of retained fetal membranes
Rew#1: Line 237-250: the sentence is maybe too long. I suggest placing at least one point, for example at line 246: "... be increased. Actually, feeding time…”
AU: It was changed: used to be increased. While, feeding time
Rew#1: Line 285: “sensitivity” maybe you intended “specificity”?
AU: It is correct. Thank you very much: while the specificity
Rew#1: Line 328: I suggest changing with: “…within 3 hours before calving. However only 15 cows were involved in the experiment [78].”
AU: It was changed: before calving. However,
Rew#1: Line 353-355: just a clarification is needed with this sentence. More in detail, the intravaginal devices are expelled from the birth canal when the sacs or fetus enter the canal itself. Vulvar magnetic sensors are not expelled in this case, but are separated when the birth canal dilates. Please, I suggest to re-arrange the phrase accordingly.
AU: It was changed: Appearing allantochorion separates the vulval lips
The intravaginal sensors will be expelled from the birth canal when the allantochorionic sac enters it.
Rew#1: Line 420-431 or 435-438: in these paragraphs the author describes both pros and cons of the different intravaginal devices for the identification of calf expulsion. Among adverse effects/defects, I suggest to mention that vaginal retention issues of the intravaginal device were observed in buffalo heifers with recurrent vaginal prolapses, as described in [Remote monitoring system as a tool for calving management in Mediterranean Buffalo heifers (Bubalus bubalis ). Emanuela Rossi, et al. Reproduction in domestic animals. Volume55, Issue12 Vol. 55, Issue 12 December 2020 Pages 1803-1807. https://doi.org/10.1111/rda.13805]. Thus, pathological conditions which can influence the functionality of vaginal tissues should be accounted for and discussed, when considering the use of those devices as calving alarms.
AU: It was added: Recurrent vaginal prolapse may abort the physical sensor, as found in a buffalo heifer, while poor carrier network coverage may also negatively affect the phone's signal quality [101].
Rew#1: Line 446-458: in this paragraph the author outlines the potential use of authomatic sensors for calving prediction; I only suggest to summarize which time interval is achieved for the prediction of calving with those systems.
AU: It was changed: not only the second stage of calving
Rew#1: Line 465: “invasive” instead of “invasion”?
AU: It is correct. Thank you very much: this is an invasive method
Reviewer 2 Report
Accuracy to predict the onset of calving in dairy farms by using different precision livestock farming devices
This is a very clearly represented and comprehensive study on the important subject of calving and its consequences on the start of the lactation.
Overall
Abstract : Unfortunately there is only 1 study [56] proving a higher rate of metritis / endometritis if calving could not be predicted. More literature to that subject ?
Page 2, lines 43-61 : As to perinatal mortality, please add as references Mee et al., Vet Clin N Am 2004 citing the 2 feet 2 hours rule and Mock et al. Theriogenology 2020 analyzing UCOD (why ?) and PCOD (how ?); both important points for this subject. Please also add the management point of having a night man watching cows on large dairies to prevent unseen cases of dystocia (Mee et al Vet Clin N Am 2004)
Chapter 3.2. Please add the results of Fadul et al Anim Reprod Sci 2017 into the text of combined sensors
Discussion : In the rev’s opinion it is important to underline cattle welfare /as done on page 9, line 429) and to go for non-invasive tools if possible and if as reliable as other tools
Small typos:
Page 1, line 41: ] missing
All over the ° are in the middle of the lines
[82] delete the a at the end of Bogdahn and Alsaaod and Steiner and the b at the end of Hüsler
Author Response
Responses to Reviewer Comments:
The author is grateful for the efforts of Reviewers in the evaluation of his manuscript. I appreciate your time spent with the review. Before answering the most important concerns, let me thank you for your valuable comments on the paper. I feel that Reviewers’ comments and recommendations were reasonable, and I tried to take them into account as far as possible while improving the manuscript. In my opinion, the activities of the reviewers have contributed significantly to the improvement of the quality of my paper. As you will see, I have made all the corrections required.
Reviewer II Comments
Accuracy to predict the onset of calving in dairy farms by using different precision livestock farming devices
This is a very clearly represented and comprehensive study on the important subject of calving and its consequences on the start of the lactation.
AU: The author would like to thank Reviewer II for finding merit in the manuscript.
Rew#2: Abstract : Unfortunately there is only 1 study [56] proving a higher rate of metritis / endometritis
AU: it was added other references: uterine diseases/clinical metritis [61,91,94]
Rew#2: Page 2, lines 43-61 : As to perinatal mortality, please add as references Mee et al., Vet Clin N Am 2004 citing the 2 feet 2 hours rule and Mock et al. Theriogenology 2020 analyzing UCOD (why ?) and PCOD (how ?); both important points for this subject. Please also add the management point of having a night man watching cows on large dairies to prevent unseen cases of dystocia (Mee et al Vet Clin N Am 2004)
AU: it was added: These figures emphasize the importance of examining the causal factors of perinatal mortality. The proximate cause-of-death (PCOD) with a non-infectious etiology is likely multifactorial. Still, most calves may die due to direct and indirect asphyxia because, in 73 to 75% of the calves that died in the perinatal period, no pathological changes were detected [15,16]. In other studies, asphyxia in calves dying perinatally was 58.3% [17] and 44.7% [18], respectively. According to recent necropsy studies, Mee [19] reported that the prevalence of anoxia was highly variable (~5 to ~80%) while the combined diagnosis rate with dystocia ranged between ~20 and ~45%. In a recent study, the ultimate cause-of-death (PCOD) with an infectious etiology was 34%, of which Coxiella burnetii was the most frequently detected pathogen [18]. This finding calls attention to the importance of monitoring the contagious etiology of perinatal mortality in those farms where the prevalence of stillbirth is high.
Rew#2: Chapter 3.2. Please add the results of Fadul et al Anim Reprod Sci 2017 into the text of combined sensors
AU: it was added: Fadul et al. [87] measured ruminating time, ruminating chews, boluses, and other activities not related to ruminating, feed intake or drinking activity (noseband sensor), and lying bouts (3-D accelerometer). Sensitivity of 88.9% and 85% and specificity of 93.3% and 74% for primiparous and multiparous dairy cows were reported to predict calving within three hours before calving.
Rew#2: Discussion : In the rev’s opinion it is important to underline cattle welfare /as done on page 9, line 429) and to go for non-invasive tools if possible and if as reliable as other tools
AU: It was added to the text:
Line 34: and improve their welfare.
Line 466: and greatly contributed to the welfare of these animals [1,36,37].
Line 500: and we can improve animal welfare [1,36,37].
Line 520: stillbirth and to improve animal welfare
Rew#2: Page 1, line 41: ] missing
AU: thank you very much: [9-11].
Rew#2: All over the ° are in the middle of the lines
AU: In the original manuscript they are at the correct places.
Rew#2: [82] delete the a at the end of Bogdahn and Alsaaod and Steiner and the b at the end of Hüsler
AU: It was corrected: Fadul, M.; Bogdahn, C.; Alsaaod, M.; Hüsler, J.; Starke, A.; Steiner, A.; Hirsbrunner, G.